Contributions of each of the four swimming strokes to elite 200-400 individual medley swimming performance in short and long course competitions

González-Ravé José María 1
Santos-Cerro Jesus jesus.scerro@uclm.es 2
González-Megía Patricia 1
Pyne David 3
1 Sports Training Laboratory, Faculty of Sport Sciences, Universidad de Castilla la Mancha , Toledo , Spain
2 Applied Economics I, Universidad de Castilla La Mancha , Toledo , Spain
3 Research Institute for Sport and Exercise, University of Canberra , Canberra , Australia
Kerhervé Hugo
Electronic publication date: 2023 Dec 15
Publication date: 2023
Volume: 11
Electronic Location ID: e16612
Received 2023 Jun 13; Accepted 2023 Nov 15
Copyright: ©2023 González-Ravé et al.
Copyright year: 2023
Copyright holder: González-Ravé et al.
License: This is an open access article distributed under the terms of the Creative Commons Attribution License, which permits unrestricted use, distribution, reproduction and adaptation in any medium and for any purpose provided that it is properly attributed. For attribution, the original author(s), title, publication source (PeerJ) and either DOI or URL of the article must be cited.
License URL: https://creativecommons.org/licenses/by/4.0/

Keywords: Performance, Data analysis, Swimming, Competitions, Tactics, Mentoring

Funding: Department of Applied Economics I (DEAI 00421I126) of the University of Castilla La Mancha, Toledo, Spain The authors received no funding for this research. The APC was funded by the Department of Applied Economics I (DEAI 00421I126) of the University of Castilla La Mancha, Toledo, Spain. The funders had no role in study design, data collection and analysis, decision to publish, or preparation of the manuscript.

==============================
Objectives

The relative contribution of each of the four strokes to performance, and whether these contributions differ substantially between short course and long course competitions is unclear. To clarify these issues the aim of this study was to assess the strokes that have more influence on the performance in the 200 and 400 m IM swimming performances of elite male and female swimmers, participating in major events: Olympic Games (OG) and World Championship (WC) in short-course and long-course from 2012 to 2021.

Methods

Data from 1,095 swimmers (501 women and 594 men) who competed in 200 and 400-m IM were obtained with a minimum level of 800 FINA points. Linear regression modelling and classification trees were employed to quantify differences between strokes and short/long course swimming.

Results

Regression analysis indicated that breaststroke (β =  − 0.191; p < 0.000) and backstroke (β =  − 0.185; p < 0.000) had a bigger effect on IM performance, with butterfly (β =  − 0.101; p < 0.000) having a lesser impact. The classification trees showed threshold performance standards in terms of 50-m times in form-stroke events must be fulfilled to attain medal-winning performances.

Conclusions

These form-stroke standards represent important milestones for designing medal-oriented training strategies for both 200 IM and 400 m IM. Achieving a medallist position in 200 and 400 m IM requires obtaining specified lap times in butterfly, breaststroke and backstroke for males and females in long-course competitions, and breaststroke and backstroke for short-course competitions. The OG presents more exigent demands of lap times in butterfly, crawl and backstroke for IM swimmers.

Introduction

The individual medley (IM) events in swimming are performed with all four competitive swimming strokes: butterfly, backstroke, breaststroke, and freestyle. The 200 m IM and 400 m IM are the most challenging middle-distance events in swimming and the complexity in their preparation gives them a special appeal for swimming coaches and researchers (Del Castillo et al., 2022; González-Ravé et al., 2022; Hermosilla et al., 2021). The importance of swimming the individual form stroke events (butterfly, backstroke, breaststroke, and freestyle over 100 m and/or 200 m) to a high standard is acknowledged by IM coaches and swimmers’ based on our long experience in competitive swimming. Elite IM swimmers usually score up to 900 or more FINA points in their main event and other competitive strokes (González-Ravé et al., 2023). Del Castillo et al. (2022) reported three different IM profiles: 200 m IM swimmers holding higher FINA points in sprint swimming, 200–400 m IM swimmers, and 400 m IM specialists holding higher FINA points in middle distance. Hermosilla et al. (2021) showed that swimmers can attain repeated performance peaking in various time frames (influenced by national and international competition calendars), and that optimal performance at a major championship should be highlighted in the periodization designed by the coach (Mujika et al., 2019).

Swimming competitions can be held in short (25 m) or long (50 m) course pools (World Aquatics, 2017). Faster times are generally achieved in short course swimming, as a function of the greater number of turns regardless of the stroke and distance performed in each event (Keskinen, Keskinen & Mero, 2007). Swimmers produce a greater propulsion and a better recovery in short-course events, yielding differences in physiological variables (heart rate, concentration blood lactate) and biomechanics (swimming efficiency) between short and long course swimming (Keskinen, Keskinen & Mero, 2007). Over a season a swimmer usually competes in both short (autumn and winter competitions) and long course (spring and summer) competitions. Performance in international events from 2000 to 2012, was ∼2% faster in short course than long course, except in long-distance events (800 and 1,500 meter freestyle) for men (Wolfrum et al., 2013; Wolfrum et al., 2014). There are also differences by sex, with male swimmers obtaining higher performances than females in IM (Nikolaidis & Knechtle, 2018).

The preparation for both the 200 m and 400 m IM events involving the combination of all four strokes creates unique energetic requirements (Pyne & Sharp, 2014) and a suitable balance in training strong and weak strokes is required. Saavedra et al. (2012) investigated the pacing strategies employed in the 200 and 400 m individual medley, and which strategy was the most determinant for the final performance as a function of sex and classification (1st to 3rd, 4th to 8th, 9th to 16th) in international competitions. Considering only the medallists, in men (200 m and 400 m IM), the backstroke was the most determinant stroke in their final performance, whereas in women, it was the backstroke (200 m) or freestyle (400 m). Males tend to adopt a positive pacing strategy in an IM, swimming relatively faster in the first half of the race, whereas females start more conservatively adopting a negative pacing strategy and swimming relatively slower in the first half of the race. According to Moser et al. (2021), for both sexes, freestyle was the fastest stroke, followed by butterfly, backstroke, and breaststroke. Bearing this in mind, the analysis of pacing and the most influential strokes on the FINA score for IM swimmers in short and long course constitute a valuable information for coaches who train elite 200 m and 400 m IM swimmers. Coaches can use this percentage distribution as a benchmark for planning training and peak performance in major competitions, providing reference values for long-course and short-course swimming pool major events for both sexes (McGibbon et al., 2020). However, the knowledge of the stroke-specific contributions to IM swimming performance, and how they differ between short or long course swimming is limited. Our interest lies in the study of the 200 m and 400 m IM performances of elite male and female swimmers, participating in major events (Olympic Games and World Championships) in short pool (25 m) and long pool (50 m) from 2012 to 2021.

Materials & Methods

An observational retrospective study of publicly available swimming competition results was conducted. Data were collected from a total of 1,095 swimmers—382 (170 women and 212 men) who competed in 200 m IM and 400 m IM in a 25 m pool, and 713 swimmers (331 women and 382 men) in 200 m IM and 400 m IM in a 50 m pool. All swimmers had a minimum of 800 FINA points in these events. According to Ruiz-Navarro et al. (2023) the threshold of 800 FINA points corresponds with swimmers that participated in international events B qualifying standards. The next level below corresponds to swimmers attending a national championship. Official results were identified, extracted, and compiled from the World Championships (25 m and 50 m) and Olympic Games (50 m only) for from the period 2012 to 2021. The data were obtained from the FINA database (https://www.worldaquatics.com/), and based on publicly available and freely available information, so no informed consent from the individuals was required.

The following variables were collected for each swimmer: season, distance (200/400 m), competition (Olympic/World Championships), FINA points, pool type (25/50 m), nationality, gender (male/female), total time, age, split time per stroke in IM event (butterfly, backstroke, breaststroke, crawl) and finishing position. All lap times of each stroke (butterfly, backstroke, breaststroke and crawl or freestyle) were converted to a Z-score via the equation: Zij=xij−x¯iσi∗100

where j = participant; i = group by gender, stroke, distance and pool. In this way we examined the effects of each partial time on the total swim time in both men and women in the 200 IM and 400 IM event, grouping them in this way by event, gender, and type of competition (Olympic and World Championships). The definition of the Z-scores considers all variables (event, gender, and type of competition, i.e., Olympic and World Championships) simultaneously. This approach allows us to create homogeneous groups characterized by the same values of these variables.

For the partial times of each of the four strokes (categorized by gender, distance, type of pool, stroke, as already mentioned), the z-score was used. A 0 value in the z-score means that the time coincides with the mean of the group, a positive value corresponds to a time above the mean of the group (slower), and a negative value to a time below the mean (faster).

Linear regression models were employed to account for FINA points as a function of the normalised partial times through the z-scores. We analyzed the significance and intensity of the size of the effect of each of the strokes on the final result of the event. Several models were built according to the type of pool (25-m or 50-m pool), gender and global. This approach was used to compare the impact of an improvement in each of the strokes on the final result of the event in terms of FINA points.

The analysis of the effect of each swimming style in the final position in the 25 and 50 metre pool by event and gender was conducted using the classification or decision tree model as part of a supervised analysis technique. This approach assigns a classification variable (in our case “class”) for which each observation (swimmer) to a group according to the following criteria: medallist (final position from 1 to 3), finalist (final position from 4 to 8), and rest (rest of positions). First, a sub-sample was selected randomly to derive each of the estimated trees (70%), and the rest of the sample (30%) used to validate and calculate the confusion matrix and precision coefficient (accuracy).

Four trees were derived, one for the case of 25 metre pool participants, one for 50 metre pool participants, one for Olympic Games participants, and the last for World Championships participants. For this purpose, the confusion matrix (from the validation sample) and precision coefficient were calculated, and graphical representation of the trees generated. The classification model can make errors between adjacent classes, i.e., some swimmers are classified by the model as medallists, finalists or rest and are not identified with the real position. Statistical analyses were performed using R software (version 4.1.2 for Windows; R Core Team, 2021). Statistical significance was accepted for p < 0.05. Only models with accuracy rated above 80% were retained.

Results

We generated five linear regression models to explain the FINA score as a function of the normalized partial times in each of the strokes by means of the z-scores. The first regression model was the global one (including all the swimmers in the sample), two models related to the 25 m and 50 m pools, and the other two models for male and female swimmers. We expected that in these models the beta coefficients would be negative, as an increase in the z-score in a given stroke (worsening of the relative time) means a lower FINA score (negative z-score values represent values below the average time). The stroke with the most negative beta coefficient would have the greatest influence on the swimmer’s FINA score. On the other hand, the most intuitive interpretation of these coefficients is the relative or comparative one. According to Table 1, in all five models all the strokes are significant, globally the models are significant, and present degrees of goodness of fit (R2) around 90%, which means an excellent fit of the models.

Table 1 FINA Point Regression Models as a function of partial style scores.

	Estimate (β)	Std. Error	t value	p value	R 2	
	 	GLOBAL	 	 	 	
(Intercept)	868.930	0.4009	2167.35			
Z_butterfly	−0.101	0.0049	−20.72			
Z_backstroke	−0.185	0.0048	−38.55	p < 0.000	92%	
Z_breaststroke	−0.191	0.0044	−43.17			
Z_crawl	−0.162	0.0043	−37.34			
 	 	SHORT COURSE	 	 	
(Intercept)	865.657	0.8181	1058.1	 		
Z_butterfly	−0.092	0.0109	−8.472			
Z_backstroke	−0.193	0.0107	−18.076	p < 0.000	89%	
Z_breaststroke	−0.171	0.0096	−17.859			
Z_crawl	−0.170	0.0090	−18.882			
 	 	LONG COURSE	 	 	
(Intercept)	870.683	0.4142	2102.1			
Z_butterfly	−0.105	0.0049	−21.68			
Z_backstroke	−0.183	0.0048	−38.34	p < 0.000	94%	
Z_breaststroke	−0.200	0.0045	−44.84			
Z_crawl	−0.158	0.0044	−35.54			
 	 	FEMALES	 	 	
(Intercept)	870.778	0.7515	1058.705	 		
Z_butterfly	−0.082	0.0089	−9.279			
Z_backstroke	−0.193	0.0090	−21.325	p < 0.000	88%	
Z_breaststroke	−0.206	0.0081	−25.308			
Z_crawl	−0.170	0.0082	−20.739			
 	 	MALES	 	 	
(Intercept)	867.370	0.3529	2457.68			
Z_butterfly	−0.119	0.0043	−27.14			
Z_backstroke	−0.173	0.0042	−41.28	p < 0.000	97%	
Z_breaststroke	−0.176	0.0039	−44.68			
Z_crawl	−0.155	0.0038	−40.81			

In all the models, based on the estimated beta coefficients, the most influential strokes on the FINA score for IM swimmers are backstroke and breaststroke, with half of this effect produced by butterfly, and crawl eliciting a smaller moderate effect. Observing the overall results by type of pool (short and long course), we identified that breaststroke is slightly more relevant in the 50-m long course swimming (β =−0.200) than 25-m pool (β =−0.171). In contrast, front crawl (freestyle) is more relevant in 25-m short course swimming than 50-m pool (β = −0.170 versus β = −0.158). In the comparison by gender, for the female swimmers the effect produced by the butterfly stroke is slightly less relevant (β = −0.082), while for the male swimmers butterfly is more important (β = −0.119).

For short course swimming, the classification accuracy of the validation sample was 73%, with the classification errors mainly in adjacent categories. On the other hand, according to the tree representation (Fig. 1), reaching the medal position is essentially due to two conditions: obtaining a breaststroke z-score lower than -67%, and a backstroke z-score lower than -52%. These conditions, in terms of absolute times in seconds specific to gender and distance (Table 2) for females in 200-m IM events are performance times faster than 36.9 s in breaststroke and 31.6 s in backstroke. For male swimmers, these limits were 32.9 and 28.2 s respectively. For females in 400-m IM events are times faster than 75.9 s in breaststroke, and 67.6 s in backstroke. For males these limits were 69.2 and 61.0 s respectively.

Figure 1 Decision trees for identifying the relative contribution of breaststroke, backstroke, and front crawl to medal-winning, finalizing, or other (rest) performances in short (upper panel) and long course (lower panel) international swimming.

*back = backstroke, fly = butterfly, breast = breaststroke ** The leaf nodes (at the bottom) correspond to the classification predicted by the tree. From the initial or root node (at the top), following each of the conditions that define the different branches, we reach each of the leaf nodes, which indicates what must be fulfilled to achieve each of the leaf nodes (finalist, medallist or rest). The percentage figure for each node corresponds to the part of the sample contained within. The three figures above this percentage correspond to the distribution of the node’s sample among the different categories (finalist, medallist and rest, respectively).

For the long course, the classification accuracy of the validation sample was 82%. According to the tree representation (Fig. 1), achieving the medal position was essentially related to three conditions: obtaining a z-score in butterfly lower than -68%, a z-score in breaststroke lower than -55%, and a z-score in backstroke lower than -31%. These conditions, in terms of absolute times in seconds according to gender and distance, are shown in Table 2. For example, for females in 200-m IM events at the threshold times were lower than 28.1 s in butterfly, 37.9 in breaststroke and 33.5 in backstroke. For males these threshold times were 25.2, 34.1 and 30.1 s respectively. For 400-m IM events these thresholds are 62.4, 79.1 and 70.8 s (females) and 56.9, 71.7 and 64.8 s (males) as shown in Table 2.

For the Olympic Games, the classification accuracy of the validation sample was 75%. According to the tree representation (Fig. 2), reaching the medal position was essentially related to obtain the following conditions, in terms of absolute times in seconds according to gender and distance (shown in Table 2). For example, for females in 200-m IM events are set at times lower than 28.5 s in butterfly, 31.1 in crawl and 32.9 in backstroke. For males these limits are 25.4, 28.4 and 29.4 s respectively. For 400-m IM events these thresholds are 63.0, 63.1 and 69.5 s (females) and 57.3, 58.7 and 63.4 s (males) as shown in Table 2.

For the World Championships, the classification accuracy of the validation sample was 79%. According to the representation of the tree, reaching the medal position was primarily related to obtaining a breaststroke z-score of less than -54% and a backstroke z-score of less than -98%. These conditions, in terms of absolute times in seconds according to gender and distance, are shown in Table 2. For example, for females in 200-m IM events and short course competitions, are set at times lower than 36.9 s in breaststroke and 31.6 in backstroke. For males these threshold values were 32.9 and 28.2 s respectively. Comparable stroke-specific thresholds for the 400-m IM events are shown in Table 2.

Discussion

This study investigated the 200 m and 400 m IM swimming race pace of elite male and female swimmers, participating in major events Olympic Games, short and long course World Championships from 2012 to 2021. This is the first work that provides a comprehensive evidenced-based set of lap times in 200 m IM and 400 m IM need to achieve success in international competitions, considering gender, event, laps, and FINA points. These details allow swimmers, coaches and sports scientists to identify the level of performance needed to achieve medallist positions in international short course (25 m pool) and long course (50 m pool) swimming competitions.

Table 2 Time thresholds in seconds according to stroke, gender, pool type and competition to achieve medallist position in 200-m IM events and 400-m IM events.

 Gender	Short course	Long course	
	200 IM	400 IM	200 IM	400 IM	
Global	
Female	breaststroke <36.9 and backstroke<31.6	breaststroke<75.9 and backstroke<67.6	butterfly<28.1 and breaststroke<37.9 and backstroke<33.5	butterfly<62,4 and breaststroke<79.1 and backstroke<70.8	
Male	breaststroke<32.9 and backstroke<28.2	breaststroke<69.2 and backstroke<61.0	butterfly<25.2 and breaststroke<34.1 and backstroke<30.1	butterfly<56.9 and breaststroke<71.7 and backstroke<64.8	
World Championships	
Female	breaststroke<36.9 and backstroke<31.6	breaststroke<75.9 and backstroke<67.6	breaststroke<37.7 and backstroke<32.9	breaststroke<79.1 and backstroke<69.5	
Male	breaststroke<32.9 and backstroke<28.2	breaststroke<69.2 and backstroke<61.0	breaststroke<34.1 and backstroke<29.6	breaststroke<71.8 and backstroke<63.6	
Olympic Games	
Female	 	 	butterfly<28.5 and crawl<31.1 and backstroke<32.9	butterfly<63.0 and crawl<63.1 and backstroke<69.5	
Male	 	 	butterfly<25.4 and crawl<28.4 and backstroke<29.4	butterfly<57.3 and crawl<58.7 and backstroke<63.4	

Figure 2 Decision trees for identifying the relative contribution of breaststroke, backstroke and front crawl to medal winning, finalise or other (rest) for the Olympic Games and World Championships.

*back = backstroke, fly = butterfly, breast = breaststroke.

Based on our results, the main outcomes were that the most influential strokes on the final IM performance are backstroke (r = 0.92; β =−0.185) and breaststroke (r = 0.92; β =−0.191) on FINA points in the whole sample. In short course swimming the most influential stroke was backstroke (r = 0.89; β =−0.193), while in long course the most important strokes were backstroke (r = 0.94; β =−0.183) and breaststroke (r = 0.94; β =−0.200). By gender, for both males and females the most influential strokes were backstroke and breaststroke (see Table 1). Butterfly stroke was the fastest irrespective of final placing or gender (Saavedra et al., 2012). The results of Saavedra et al. (2012) also confirmed that male swimmers adopted faster swimming speeds over the first half of 200 m and 400 m IM, while female swimmers were able to finish the race faster in relative terms than men. This disparity could explain the lack of signification in freestyle among males (McGibbon et al., 2018). The explanation for this gender effect is unclear but might relate to lower accumulation of lactic acid in these events in female swimmers. According to our results, an effective pacing strategy focusing on backstroke and breaststroke is also necessary given that improvements in specific lap times are associated with substantial improvements in final time for 100 to 400 m swimming events (McGibbon et al., 2018; Robertson et al., 2009). Coaches should seek to improve the slowest stroke to maintain a competitive race position.

The decision tree analysis provides detailed lap-time information, allowing us to identify the likelihood of an elite swimmer achieving a medallist position, becoming a finalist, or achieving another result. According to results shown in Table 2, achieving a medallist position in 200 m and 400 m IM requires obtaining specific lap times in butterfly, breaststroke and backstroke for males and females in long course competitions, and breaststroke and backstroke for short course competitions. It is worth pointing out that three strokes (butterfly, breaststroke, backstroke) are relevant for achieving medallist positions in long course World Championships and Olympic Games, while breaststroke and backstroke are the priority in short course World Championships. These results differ from the study of Saavedra et al. (2012) who affirmed that backstroke was the stroke that most determined their final performance, whereas, in women, it was the backstroke (200 m) or freestyle (400 m). We should consider the work of Saavedra et al. (2012) analysed the period between 2000 and 2011 (eleven years before our data was collected). The estimates were made on data gathered from the Olympic Games, World Championships, European Championships, Commonwealth Games, Pan-pacific games and EEUU trials. The evolution and development of swimming in the 2000–2011 (Saavedra et al., 2012) is different to 2012–2021 (this research). Also, we acknowledge that specific issues such the prohibition of the high-tech swimsuit in 2009, or the pandemic lock down in 2020 could also affect the results. However, our study period commenced after the 2009 high-tech swimsuit period. Employing different statistical analyses and the use of Z-score, instead of a two-way analysis of variance may have also influenced the interpretation of selected results.

Clearly the 200 m and 400 m IM performances are faster in short course swimming. These results are consistent with studies showing that that the pool length has a strong effect on blood lactate concentration and heart rate with greater swimming velocity in the short course pool (Keskinen, Keskinen & Mero, 2007; Lowensteyn et al., 1994; Holfelder, Brown & Bubeck, 2013). It is well known that male swimmers (athletes) typically exhibit higher lactate values than female swimmers (Ferreira et al., 2016; Holfelder, Brown & Bubeck, 2013). The best lap times are obtained in short course World Championships compared with Olympics and long course World Championships where the frequency of turning likely plays an important role in regulating physiological response to incremental intermittent swimming exercise (Keskinen, Keskinen & Mero, 2007). Each turn provides (brief) recovery time moderating increases in lactate production and increasing in lactate clearance in the upper body and arm muscles (Keskinen, Keskinen & Mero, 2007). Coaches should pay attention to the periodization of training for an elite 400 m IM swimmer, where the first performance peak is typically achieved in December when competing in 25 m pool according to the World Aquatics calendar (González-Ravé et al., 2022).

The major competition over the season is habitually held in July-August where Olympic Games or long course World Championships are scheduled. Elite 200 and 400 m IM swimmers competing at international events (Olympic Games and world Championship) had the best lap times in the Olympic events. The explanation may be the long course World Championships competitions typically have more athletes, so the level of competition is quite different compared with Olympic Games. The Olympic Games takes place every 4 years, and FINA World Championships, which are held in pre- and post-Olympic years The Olympics are considered the pinnacle of any athlete’s career, and the most elite swimmers focus their peak performance on this competition. The Olympic Games is a measurable test of a nation’s sporting power and medals won are the object of intense scrutiny before and after every Olympiad (Seiler, 2013).

These results are limited to comparison of FINA points lap times between IM events and other events that swimmers habitually compete over the season (i.e., 100 m for 200 m IM tests and 200 m tests for 400 IM). However, threshold values for form-stroke event performance times (standards) can be used immediately by swimming coaches for planning of IM-specific training sets, and evaluating progression towards training and competition goals. Future studies should investigate more detailed relationships between anthropometric and training characteristics of IM swimmers, and previous FINA points. This approach should establish new lines of research for improving the training and competitive performance of IM swimmers. Furthermore, the statistical analysis conducted involved two significant factors: the type of competition (World Championships and Olympic Games) and the length of pool (25 m and 50 m), which encompass the other two variables (distance and gender). To fully explore the interactions of these four variables, a larger sample size would be necessary.

Conclusions

Three swimming strokes (butterfly, breaststroke, backstroke) are important for achieving medallist positions in IM events in long course World Championships and Olympic Games. Breaststroke and backstroke are the priority in short course World Championships, while our results also established as breaststroke and butterfly stroke have a greater and lesser influence on the final time, respectively. The Olympic Games present more exigent demands of lap times in butterfly, crawl, and backstroke (it takes place every four years and has more stringent classification times) than the World Championships. Achieving a medal position in World Championships was primarily related to obtaining target or threshold times in breaststroke and backstroke to increase the likelihood of competitive success.

Supplemental Information

Supplemental Information 1 Raw data

The information is available from World Aquatics (https://www.worldaquatics.com/results?year=2023&month=latest&disciplines=SW). Data collection was conducted for each of the years under study, for each type of event-distance and gender, as well as for each type of competition (Olympic Games and World Championships). This data collection process was exceptionally labor-intensive due to the necessity of extracting data from numerous tables found on the website.

Click here for additional data file.

Additional Information and Declarations

Competing Interests

Author Contributions

Data Availability

The authors declare there are no competing interests.

José María González-Ravé conceived and designed the experiments, authored or reviewed drafts of the article, and approved the final draft.

Jesus Santos-Cerro conceived and designed the experiments, analyzed the data, prepared figures and/or tables, authored or reviewed drafts of the article, and approved the final draft.

Patricia González-Megía conceived and designed the experiments, performed the experiments, authored or reviewed drafts of the article, and approved the final draft.

David Pyne conceived and designed the experiments, authored or reviewed drafts of the article, and approved the final draft.

The following information was supplied regarding data availability:

The data is available in the Supplemental File and at World Aquatics (https://www.worldaquatics.com/results?year=2023&month=latest&disciplines=SW).

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
