# Peer review of "Contributions of each of the four swimming strokes to elite 200-400 individual medley swimming performance in short and long course competitions"

_PeerJ, doi:10.7717/peerj.16612_

## Round 0.1 · original submission · Major Revisions

The reviewers all noted the importance of the work for the field and the amount of work performed.

- The main comments pertain to the need to strengthen the rationale of the article, making it more readily understandable and plainly visible.

- There were a few comments honing on to the grouping of data and their associated differences (sex for instance).

- Some methodological clarifications and further justification of methods are required.

- Since no physiological measures were performed, the reviewers noted the need to use caution in the discussion when explaining potential differences (eg, lactate accumulation), potentially by linking specific research on the topic.

I recommend major revisions given that many pointers were kindly provided by the reviewers, but there were no major scope or conceptual limitations.

Reviewer 1 ·

Basic reporting

Thank you for the opportunity to participate in the review and evaluation process regarding the paper “Contributions of each of the four swimming strokes
to elite 200-400 individual medley swimming performance in short and long course competitions” submitted to the PeerJ.

Although respecting the work amount authors had been dedicated into their study, the manuscript in current version is not convincing enough regarding what knowledge gap being filled

Experimental design

Your must important issue is the lack of presenting the study’ unprecedented contribution of the study to the area. The authors justify the importance of the study by the limitation of information regarding stroke-specific contributions to individual medley swimming performance in short or long course swimming, however Saavedra at el. (2012) previously studied the subject... what is the difference between this work for Saavedra et al. (2012)? What is the literary gap that the proposal intends to cover?

Methods
Sample characterization information was not given, what can directly influence the results (height and wingspan, for example). This limitation must be recognized.

Line 111-114
Some information collected was not presented in the study.

Validity of the findings

Lines 262-264
“The explanation for this gender effect is unclear but might relate to lower accumulation of lactic acid in these events in female swimmers” this explanation is speculative... needs to be deepened and grounded in the literature. Would there be any other social, physiological or psychological argument to explain this strategy difference between the genders?

Lines 277-281
Is the difference between period used in this study and that of Saavedra et al. (2012) is the only explanation that the authors can attribute to the differences in the data results? Do the authors suggest that the stroke-specific contributions to individual medley swimming performance changes over the years? Do the results then only include a finding for the analyzed moment and future competitions may not correspond with the indicated results? If so, the time period is a limitation for future applications…

Additional comments

No additional comment

·

Basic reporting

The authors aimed to assess the most determinant swimming strokes in achieving a medal in 200 m and 400 m IM events, either in short course and long course both in world championships and Olympic games. A brief research in the literature shows a lack of evidence in this topic, thus room for further investigation. Despite no practical data collection was needed, the amount of data gathered for this research must have been enormous and, adding up the number of swimmers taken into account, the robustness of the data is a plus in this research.

Figure resolution should be improved, as, in this version of the manuscript, the quality is somewhat low.

No hypotheses were formulated.

Experimental design

Some major concerns however, are raised on the statistical analysis. Not necessarily on the performed tests, but on the way the data was grouped. More details are given during the specific comments but, globally, the authors need to explain why the analyses (both in decision trees and regression of Z-scores) were made controlling for one variable at the time (e.g. controlling for pool length but not controlling for sex or swimming distance) instead of all variables at once, given that the other variables can probably bias the results.

Validity of the findings

The validity of the findings are depending on further explanation on the way the statistical tests were conducted.

Conclusion is well stated and in accordance with the results presented.

Additional comments

Specific comments:

L22-26: The first two sentences are not in accordance with the title "objectives". Please either change the title or rephrase so the sentences match the title.
L26: The authors state the aim was to “analyze”, but the very title says otherwise. In my perspective, the aim was, among others, to assess the strokes that have more influence on the performance in various IM swimming events. Please change accordingly.
L34-35: This sentence seems too vague. If the authors are not word-short, please include a brief description of the results. If the number of words is counted, please just rephrase to, at least, give a glimpse of the results. I’d suggest something like “The classification trees showed that different conditions must be fulfilled to …”. Nevertheless, a brief description of the results would be preferable.
L35-37: The authors just started a new subtopic, thus "these conditions" are actually uknown to the reader. Please rephrase for a better understanding.

L37-39: Regarding long course events, according to table 2, this seem to be true only in OG long course events. Please explain me the rationale or rephrase.
L46-47: Are there any other competitive swimming strokes? Please consider changing to "...with all four competitive swimming strokes".
L47: Please remove the dash "-" between the numbers and the units (400 m) throughout the manuscript.

L50-52: I'm sorry but I wasn't able to understand the meaning of this sentence. Please correct the english if isn't right or kindly explain its meaning.
L53-55 Please insert the subject in this part of the sentence, which I believe to be "swimmers". "...200 m IM swimmers..." if i'm correct.

L53-55 Please space the “m” and the “IM”.
L64-66 So the reader can identify straight away where the differences occur, please change to "better recovery in short-course events, yelding...".

L64-66 The use of the word "between" implies the comparison of physiologic vs biomechanic differences, while the comparison is actually between 25 m or 50 m pools. Thus, please change to "differences in physiological..."

L68-69: "in 25 and 50 m. (pool) competitions" is redundant. The phrase "Performances in international events from 2000...." is clear enough. Please change it accordingly.

L68-69: Isn’t there any literature on this topic earlier than 2012? This is because the present study focus on >2012 but this rationale is from <2012.
L71: Please chance to “performances”.
L74: Please change to "...200 m and 400 m IM events involving..."
L75: Please delete the comma before the "and". Also, please revise this misusage of the comma troughout the manuscript (e.g. L77).
L78: The inclusion of "(men and women)" is redundant. Please consider deleting it.

L82: Please change to "...IM event" throughout the manuscript whenever the "IM" reports to an event being swum.

L84-85: 1- Are the authors still talking about female events? As in the previous sentence there was a division that ended in the analysis of female swimming performance, please indicate in this sentence weather the information is still about the female, male or overall performance.
2- Are the authors talking about 200 m, 400 m or both events? Please clarify.

L87: Please insert the subject in this part of the sentence, which I believe to be "swimmers". "..400 m IM swimmers..." if i'm correct.

L111-114: Please separate the variables using a comma instead of a semicolon,

L130: Possible change to ".. and size of the effect".

L145: Given the differences between WC and OG, between short- and long-course events, between 200m and 400m IM and between sexes, what is the rationale behind an analysis controlling for only one out of 4 variables? E.g. One tree controling for 50 m pool but pooling both WC and OG, both 200 m and 400 m IM and both sexes.
Wouldn't the analyses be more accurate if there were trees for every condition?
Or another statistical analysis that granted the lack of difference between conditions so all variables could be pooled safely?
Please explain further.

L148-150: Please explain further. There could be errors in the model? Please explain how these errors were identified and handled.

L158: The authors sometimes use the word "general" and other times "global". Please revise the manuscript and improve consistency.

L157-159: Given the differences between short- and long-course events, between 200m and 400m and between sexes, what are the insights this "global" analysis can bring that the "grouping" analyses cannot? I.e., why include it?
Regarding this comment I'd need further explanation on the use of z-score statistics. Does the transformation into z-scores already take into account the differences between events, sex and pool lenght? If so, this should be clearer in the statistical analysis subsection. If not, it would be of interest to generate one table for each combination of variables (e.g., table 1 - OG, 50m, 200 IM, female swimmers; table 2 - OG, 50m, 400 IM, female swimmers; and so on).

L162: Please explain the correct terminology: is it beta coefficient or just beta? I’d suggest the authors to use always the same terminology for consistency’s sake.

L246: In my opinion, "allow" would be more suitable than "permit". Please consider changing it.

L246-248: Please change to "...the level of performance needed to achieve..."

L259-262: From "because" onward, the information is redundant with what is stated in the first part of the sentence (L 256-259). Please consider deleting it or change it to reflect an improvement in the rationale instead of repeated information.

L262-264: Please explain this rationale.

L270-271: This sentence is somewhat confusing. Please rephrase for a better understanding.

L273: Please change to "specific".

L281-283: Why this differences between Saavedra's study and the present study (namely the years and competitions) affect the results? Please explain further.

L285-288 - The HR and BLa was not measured nor assessed, so the wording "indicating" seems inapropriate. In fact, this whole sentence seems mismatched. The rationale provided in subsequent lines (L288-293) is clear enough to illustrate the difference in speed between short and long course events. Please consider rephrasing or deleting the frase (L285-288) and possibly include the references in the lines below (L288-293).

L286: Something went wrong here, please correct.

L293-296: When reading, it seems that the authors are trying to warn coaches about the difference in differences in speed between shourt and long course. However, this sentence seems to be about a new topic, namely periodization, such as the next paragraph. Please consider beggining the paragraph here (L293) instead of in line 298.

L299: Please change to "IM swimmers".

L299-303 The authors say the times differ. However, not only time but also the determinant strokes differ. Please care to give a possible explanation for this phenomenon.

L304: In “post-Olympic years The Olympics” possibly a full stop is missing.

·

Basic reporting

Congratulations to the authors for the work developed so far.

Please improve the quality of your Figures
No English language issues were detected

Experimental design

L. 91 – 92: Please add 2 – 3 lines to further develop your research question. In addition, adding a research hypothesis will improve to justify the outcomes of your study better.

Please provide a reference for the approach used to analyze the data.

L. 35 – 37: “These conditions represent important milestones for designing medal-oriented training strategies for both 200 IM and 400-m 37 IM”. Please discuss further this statement in the Discussion part. This could serve as practical application information for swim coaches.

Validity of the findings

The main strength is the novelty of the data presented.

Additional comments

L. 250 – 254: It is not necessary to repeat the statistical values in the Discussion part.

---

## Round 0.2 · Minor Revisions

Thanks for provided a detailed response and amended manuscript. One reviewer still has minor comments which warrant a response from you. These have the opportunity to further strengthen your rationale and the validity of your findings

Reviewer 1 ·

Basic reporting

No comment

Experimental design

No comment

Validity of the findings

No comment

·

Basic reporting

I thank to the authors for taking time to answer my questions. All my major concerns have been addressed and only minor revisions are required.

Experimental design

Good as is.

Validity of the findings

Good as is.

Additional comments

Specific comments in the attached pdf file.

·

Basic reporting

All my concerns have been successfully addressed. Congratulations to the authors for this work.

Experimental design

All my concerns have been successfully addressed. Congratulations to the authors for this work.

Validity of the findings

All my concerns have been successfully addressed. Congratulations to the authors for this work.

Additional comments

All my concerns have been successfully addressed. Congratulations to the authors for this work.

---

## Round 0.3 · accepted · Accept

The authors have addressed all the reviewers' comments in a constructive manner. I have reviewed this last round of comments and am happy to consider the manuscript ready for publication.

·

Basic reporting

Nothing to add.

Experimental design

Nothing to add.

Validity of the findings

Nothing to add.

Additional comments

All my concerns have been addressed. Congratulations to the authors on the quality of this manuscript.